# Microbial Biopesticides to Control Whiteflies in Eggplant *Solanum melongena*, in Greenhouse

**DOI:** 10.3390/insects16060578

**Published:** 2025-05-30

**Authors:** Héctor Cabrera-Mireles, Magali Jiménez-Jiménez, Juan Ruiz-Ramírez, Félix David Murillo-Cuevas, Jacel Adame-García, Jorge Jiménez-Zilli, Andrés Vásquez Hernández, Rubén Uriel Herrera-Bonilla

**Affiliations:** 1Instituto Nacional de Investigaciones Forestales, Agrícolas y Pecuarias, Campo Experimental Cotaxtla, Cotaxtla 94284, Mexico; vasquez.andres@inifap.gob.mx (A.V.H.); rubenurielherrbon@gmail.com (R.U.H.-B.); 2Colegio de Postgraduados, Campus Veracruz, Veracruz 91690, Mexico; jimenez.magali@colpos.mx; 3Facultad de Economía, Universidad Veracruzana, Xalapa 91000, Mexico; 4Instituto Tecnológico de Úrsulo Galván, Instituto Tecnológico Nacional de México, Ursulo Galván 91667, Mexico; felix.mc@ugalvan.tecnm.mx (F.D.M.-C.); dra.adame.garcia@gmail.com (J.A.-G.); 5Laboratorio de Biotecnología Andreb, Huatusco 94100, Mexico; jjzmx@yahoo.com.mx

**Keywords:** entomopathogens, *Cordyceps javanica*, *Beauveria bassiana*, whiteflies

## Abstract

The whitefly is a persistent pest that attacks vegetables both in greenhouses and in open fields. This pest is the main barrier to eggplant, *Solanum melongena*, production. To combat the whitefly, there is an excessive use of insecticides, and often toxic combinations of highly contaminating pesticides are applied, which are harmful to both field workers and consumers. The use of entomopathogenic fungi is a viable alternative for controlling the whitefly; however, in Mexico, there is a lack of research supporting it. Therefore, this study was established with the aim of comparing two strains of entomopathogenic fungi to the leading market insecticide for whitefly control in greenhouse eggplant cultivation. The evaluated strains proved to be equally effective as the commercial insecticide, which makes them suitable for use as microbial control agents against whiteflies. The use of the entomopathogenic fungi *Cordyceps javanica* and *Beauveria bassiana* can reduce environmental contamination and protect both applicators and consumers.

## 1. Introduction

The production of vegetables is experiencing a growth trend primarily influenced by changes in lifestyles and consumption patterns, characterized by a tendency to consume fresh and quality vegetables [1,2]. In 2023, Mexico achieved a national production of 90,000 t and stood out with yields higher than those of the world’s leading producer. Exports averaged 74,527 t annually, with an average value of 50 million dollars. The United States is the world’s leading importer, a position that it has maintained in recent years. A determining factor has been the continuous growth of its demand for eggplant [3].

The whitefly is among the main pests of crops, especially solanaceous and cucurbitaceous plants [4,5]. In Mexico, it is common to find populations of whiteflies (*Bemisia tabaci* Genn and *Trialeurodes vaporariorum* (Westwood)) in fields of beans, cucumbers, watermelons, tomatoes, peppers, and eggplants, although populations of *T. vaporariorum* are generally larger. *T. vaporariorum* affects the vegetative growth, transmits viruses that cause diseases, and promotes the development of sooty mold on leaves and fruits [6]. Its feeding also causes weakening due to nutrient extraction [7]. The control of this pest has been based on chemical products, which tend to be abused or improperly applied. Their massive use has led to an increase in the pest’s resistance to different groups of insecticides and contaminates the environment [8].

The implementation of sustainable agriculture for pest control using biopesticides will allow for the rational management and use of natural resources, food security, and the reduction in environmental pollution [9,10]. The development of new biopesticides will stimulate the modernization of agriculture and undoubtedly allow for the gradual replacement of an indefinite number of chemical pesticides [11,12]. Modern vegetable production occurs in greenhouses covered with an anti-aphid mesh with a plastic cover placed under the roof, which reduces UV irradiation by 10 to 50x [13,14]. This system enhances the yield and quality of vegetables and reduces the presence of pests and diseases [14].

Mycoinsecticides (products formulated with entomopathogenic fungi) constitute a small fraction of biopesticides [15,16]. There are more than 700 species in 100 genera worldwide, but few are studied intensively. Entomopathogenic fungi are the most important microorganisms that infect sucking insects such as aphids, whiteflies, scales, leafhoppers, and stink bugs [15,17,18]. Entomopathogenic fungi belonging to the genus *Cordyceps* (=*Isaria*) (Frieder. & Bally) Kepler and *Beauveria* (Bals.) Vuill (Hypocreales: Cordycipitaceae) [8] are promising candidates for the microbial control of insect pests [19]. They are cheaper in the long run, show fewer residual effects than chemical pesticides, and are able to overcome the problem of resistance [20]. Most experiments on the effects of microbial control agents on non-target organisms resulted in overall compatibility of the latter with the biological control agents, but some results are controversial [21]; on the other hand, Abonyo et al. [22] reported that when evaluating *M. anisopliae*, they did not observe direct effects on beneficial fauna.

In Mexico, there are few studies focused on the use of entomopathogenic fungi for the microbial control of the whitefly *T. vaporariorum*; various botanical and microbial products have been evaluated in greenhouses, such as neem formulations [23,24] as well as neem oil + *M. anisopliae* and neem oil + *C. javanica* in tomatoes and *B. bassiana* and *M. anisopliae* in pepper crops [25]. Of the 15 laboratories in Mexico that formulate agricultural insecticides based on entomopathogenic fungi, only three are authorized to use their products against *Trialeurodes* sp. for eggplant cultivation, and only one for testing against *T*. spp. [16]. In this context, the objective of the present research was to evaluate the efficiency of biopesticides based on entomopathogenic fungi against whiteflies in eggplant cultivation in a greenhouse.

## 2. Materials and Methods

Treatments. The treatments evaluated were the biopesticides *Cordyceps javanica* from the collection of entomopathogenic fungi of CNRCB-SENASICA (Colima, Mexico), Ref. [26] and *Beauveria bassiana* (collected on *Hypothenemus hampei* (Ferrari, 1867) on coffee fruits in Huatusco, Veracruz, Mexico; unpublished). Both mycoinsecticides were formulated at doses of 2.5 mL/L at a concentration of 1 × 10^8^ spores/mL, while the commercial insecticide, i.e., Spirotetramat (Movento^®^ Bayer CropScience, Mexico City, México), was administered at a dose of 1.25 mL/L, and the control consisted of no treatment application. To estimate the percentage of germination (viability) of the conidia of the biopesticides evaluated, the methodology described by Gómez [27] and Vélez [28] was followed. Dilutions were prepared in a final volume of 15 mL of a 0.1% Inex solution used as a surfactant, at a final concentration of approximately 9–12 × 10^6^ conidia/mL. Subsequently, 5 μL of the prepared dilution was taken and inoculated in Petri dishes with potato, dextrose, agar (PDA), using a sterile angled rod. The inoculated dishes were incubated at 25 ± 2 °C for 24 h. After this incubation time in a laminar flow chamber, a portion of agar of about 1 cm^2^ was cut and placed on a slide, a drop of lactophenol blue was added, and the sample was covered with a coverslip. In total, 5 samples were obtained per plate, and at least 100 conidia per sample were recorded. Germinating and non-germinating spores were counted (germinated conidia were considered those whose germination tube was twice as large as the diameter of the conidia), and finally, the average of the 5 readings was obtained, and the percentage of germinated and non-germinated conidia was calculated, using the following formula: $$\%\;Germination=\left(\frac{a}{a+b}\right)*100$$
where a = number of germinated conidia, and b = number of ungerminated conidia.

Temperature and relative humidity were recorded during the study period. The final biopesticide formulations were adjusted to 80% of viable conidia. For the application of the treatments, the pH of water was initially adjusted to 5.5 using the commercial acidifying and buffering agent DAP-PLUS at 1.2 mL/L of water, complemented with the non-ionic adherent INEX-A at 1 mL/L of water. The products were applied using an electric backpack sprayer with a fine droplet size to achieve better coverage of the upper and lower leaf surfaces.

Experimental design. The experimental plots were in a cenital greenhouse that allowed for the exit of hot air, with a plastic cover with UV blockers placed under the roof, measuring 12 m in width and 50 m in length (600 m^2^); it had a dome covered with transparent plastic and sides made of a white anti-aphid mesh. Two planting beds were prepared, measuring 1.0 m in width, 46.5 m in length, and 40 cm in height, with a 1.0 m separation between the beds, covered with black-and-white plastic mulch. The plots were organized according to a completely randomized design, with 16 repetitions per treatment. The experimental plot for each treatment had 6 plants. Each block had four treatments (24 plants).

Sampling design. From each experimental plot, two central plants were sampled; two leaves were taken from each plant, one from the upper part, and the other from the middle part, recording the presence of whiteflies by their developmental stage, at each sampling date. The first count was of the adults present on the underside of the leaves; subsequently, each leaf was cut, placed individually in properly labeled transparent polypropylene bags, and transported to the laboratory. In the laboratory, the counting of eggs and live nymphs was performed with the help of a microscope. A paper mold with a sample space of 1.44 cm^2^ (1.2 × 1.2 cm) was placed on the underside of the leaf over the second vein. The first sampling was conducted in the morning before the application of the treatments, and the following samplings were performed 1, 3, 5, 7, and 14 days after treatment application (DAA), based on the methodology used by different authors [29,30,31].

Application time. As we worked with natural populations that invaded the eggplant crop, continuous monitoring was carried out to know the number of plants with the presence of adults, nymphs, and eggs. When 60% of the plants were infested, it was decided to apply the biopesticides. The application was carried out in the afternoon, on the same day as the first sampling. Each experimental plot was surrounded by a plastic barrier for isolation during the application. In this study, 80 mL of each treatment was applied per experimental plot.

Statistical design. A completely randomized design with four treatments and 16 repetitions was used. The response variables were the numbers of eggs, nymphs, and live adults. The methodology used for the evaluation of the entomopathogens considered sampling the surviving organisms and estimating the mortality that occurred [31,32,33,34]. The data from the samplings were transformed to efficiency (%) using the Henderson and Tilton formula [35]:$$\mathrm{EEfficiency}\%=\left(1-\frac{\left(n\;in\;Co\;before\;treatment*n\;in\;T\;after\;treatment\right)}{\left(n\;in\;Co\;after\;treatment*n\;in\;T\;before\;treatment\right)}\right)*100$$
where n = number of insects, T = treated, and Co = control.

As the assumptions of the analysis of variance were not met [36,37], the non-parametric Kruskal–Wallis test was applied independently for each DAA using Statistica 7.0 software [38] to test the hypothesis that the medians of the four treatments were statistically equal; multiple comparison of the medians of the ranges of groups with a significance level of 5% was also selected, as described in [39], and the electronic manual of Statistica (version 7) software was used [38].

The post-hoc tests were complemented with a descriptive analysis by means of a graph of the confidence interval of the mean for each BP performed for the variables percentages of mortality of the flies in the egg, pupal, and adult stages, in order to show the marked differences between BPs that caused the significance obtained in the Kruskal–Wallis test. For this reason, it was not necessary to calculate the size of the non-parametric effect when comparing two groups of observations with the Cliff Delta statistic, which does not perform a hypothesis test and only serves to indicate whether two groups or treatments are equal (in this case, the value is approximately equal to zero) or different (in this case, it is equal to 1 or −1) [40].

## 3. Results

The fluctuation in the number of whiteflies estimated during the study period in the experimental batches assigned for each treatment showed that, in the initial sampling, the populations were similar in all treatments; the populations estimated in the blank treatment sample showed increases from DAA1 and continued to be the highest in the study. In contrast, populations in the lots treated with the entomopathogens and the insecticide tended to decline in a similar manner, as shown by the mean confidence intervals (Figure 1). The average temperature recorded during the study period was 27.5 ± 0.26 °C, with a maximum value of 37.9 and a minimum of 21.8 °C, and the relative humidity was 69.4 ± 0.53%, with a maximum value of 85.6 and a minimum of 59.4%.

Statistically significant differences were recorded between the efficiencies of the treatments on eggs (H = 400.50, *p* < 0.0001), nymphs (H = 315.97, *p* < 0.0001), and adults (H = 209.67, *p* < 0.0001). Statistically significant differences were observed between the pesticides in the control of the egg, nymph, and adult stages; however, no significant difference was found between the biopesticides and the commercial insecticide (Figure 2).

Considering the days after application (DAA), a statistical difference was recorded between the pesticides and the control on each of the sampling dates (Figure 3; means with the same letter are not significantly different (*p* > 0.05)).

Random samples were taken from the whiteflies killed, which were inoculated in agar dextrose Sabouraud culture medium and incubated at 25 °C to confirm that they were infected by the applied entomopathogenic fungi.

## 4. Discussion

The efficiency values of the biopesticides obtained in this study for the egg stage ranged from 93% to 97%, for nymphs from 93% to 95%, and for adults from 73% to 89%. This aligns with the findings of García-Gutiérrez and González-Maldonado [41], who evaluated bioinsecticides based on entomopathogenic fungi for managing the whitefly *Bemisia tabaci* on lettuce (*Lactuca sativa* L.), radish (*Raphanus sativus*), onion (*Allium cepa* L.), cabbage (*Brassica oleracea* var. capitata), potato (*Solanum tuberosum* L.), and cilantro (*Coriandrum sativum* L.) and showed that all isolates caused over 80% mortality at 72 h compared to the control. Our results differ from those reported by Laeun et al. [42], who stated that the rate of mycosis in early developmental stages was much higher than in eggs; in fact, in our work, the efficiency of the biopesticides against eggs was between 93% and 97%.

The efficiency observed for *C. javanica* was higher than that reported by Scorsetti et al. [43], who conducted pathogenicity tests against nymphs of *T. vaporariorum* and reported a mortality rate ranging from 26.6% to 76.6% seven days post-infection caused by *Lecanicillium lecanii* (Zimmerm.) Zare and W. Gams, *L. muscarium* (Petch) Zare and W. Gams, *L. longisporum* (Petch) Zare and W. Gams, *Cordyceps fumosorosea* Wize, and *C. javanica*. Additionally, our results showed an efficiency exceeding 89% from the day after application, which remained consistent until day 14. This contrasts with D’Alessandro [44], who reported that the application of *I. fumosorosea* in tomato and eggplant plantations initiated infection in whitefly individuals starting from day four, although it coincides with the observation that the infection continued for up to 15 days post-application. Our results also differ from those of Cabanillas and Jones [45], who indicated that the mean time to death was shorter for the second instar (3 days) than for the third instar (4 days) when exposed to 1000 spores/mm^2^ of *Cordyceps* sp., while mycosis in adult whiteflies became evident after late infections of *Bemisia tabaci* caused by this fungus.

The results obtained in this work with *B. bassiana* surpass those reported by various authors, such as Orozco-Santos et al. [46], who observed that *B. bassiana* reduced the population of *B. argentifolii* in cantaloupe cultivation under field conditions compared to the control, achieving nymph control rates between 68.9% and 81.7%. Quesada-Moraga et al. [47] reported that 25 isolates of *B. bassiana* were pathogenic to both species of whitefly, while the mortality rates varied from 3% to 85%. Kim et al. [48] noted that the insecticidal activity of *B. bassiana* M130 was 55.2% in greenhouses, 84.6% in Petri dish trials, and 45.3% in field tests. The highest efficiency of the evaluated entomopathogens occurred seven days post-application [32]. Paradza et al. [49] reported that *B. bassiana* was pathogenic to *T. vaporariorum*, with mortality rates of 45–93% for adults and 24–89% for nymphs, while Aiuchi et al. [50] reported a maximum mortality of 65.2% when evaluating commercial strains of *Lecanicillium* spp. against nymphs of *T. vaporariorum*. Javar et al. [51] reported a maximum mortality range of 74% to 89% for nymphs of *T. vaporariorum* when assessing the efficiency of *B. bassiana* cultivated on different substrates and for different storage times. Gebremariam et al. [52] evaluated strains of *B. bassiana* and *M. anisopliae* against *T. vaporariorum* and recorded mortality rates from 59% to 95% for adults and from 60% to 100% for nymphs, as well as a lethal time 50% for nymphs ranging from 3 to 8 days. Gebremariam et al. [53] achieved mortality rates of 84% to 87% in nymphs and 62% to 82% in adults of *T. vaporariorum*. Komagata et al. [54] reported that *B. bassiana* achieved an efficiency that ranged from 41.2% to 82.2%. In evaluations against *Thrips tabaci*, mortality values ranging from 6% to 79% were recorded in samples taken 3, 5, 7, and 10 days after treatment application [29].

In this study, the experimental data demonstrated that the products evaluated in their reported formulation are efficient in controlling the whitefly pest. This knowledge is unprecedented and opens great possibilities for their mass application. This is the first report of their efficiency against whiteflies in vegetables, and it was shown that they are as effective as the best insecticide recommended against whiteflies in Mexico. Probably, in a subsequent study, we will seek to determine their optimal concentrations, but in the meantime, in their current formulation, they can already be recommended. Both entomopathogens are of national origin and were obtained from national collections, which gives them an added value for their probable better adaptability to our environments, but this should be part of a subsequent study.

## 5. Conclusions

The biopesticides evaluated demonstrated high efficiency in controlling populations of whitefly in an eggplant cultivation in greenhouse.The efficiency of *C. javanica* and *B. bassiana* was similar to that of the commercial insecticide Spirotetramat for controlling the developmental stages of the whitefly.The biopesticides showed high efficiency in controlling the whitefly from the first day of evaluation and maintained this effectiveness for the full 14 days of the study.

## Figures and Tables

**Figure 1 insects-16-00578-f001:**
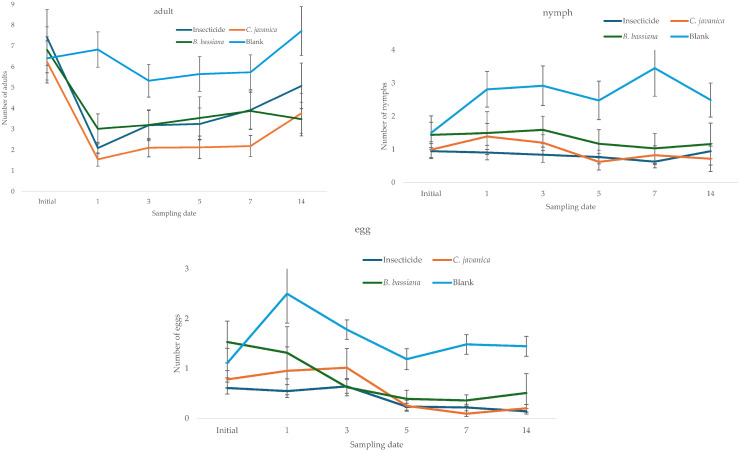
Fluctuations in the numbers of adults, nymphs, and eggs of whitefly estimated at each sampling date.

**Figure 2 insects-16-00578-f002:**
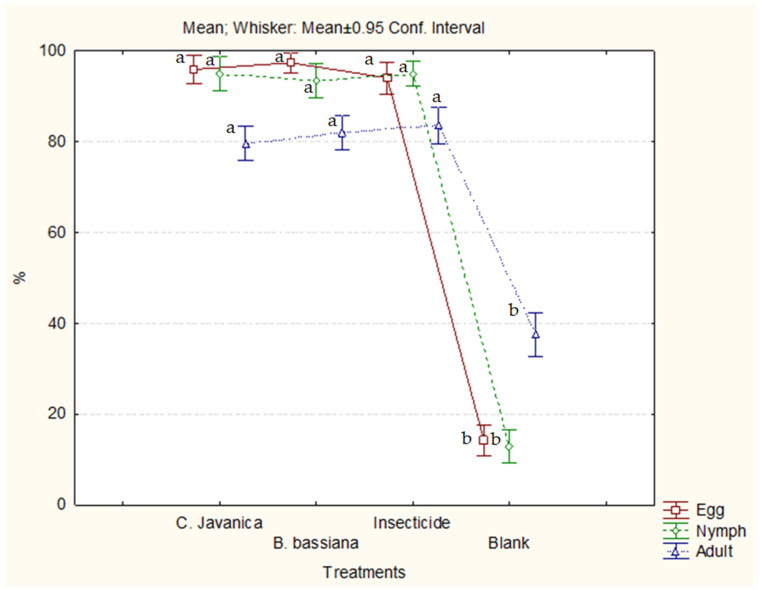
Efficiency of treatments in controlling eggs, nymphs, and adults of whitefly on eggplants in a greenhouse. Means with the same letter are not significantly different (*p* > 0.05).

**Figure 3 insects-16-00578-f003:**
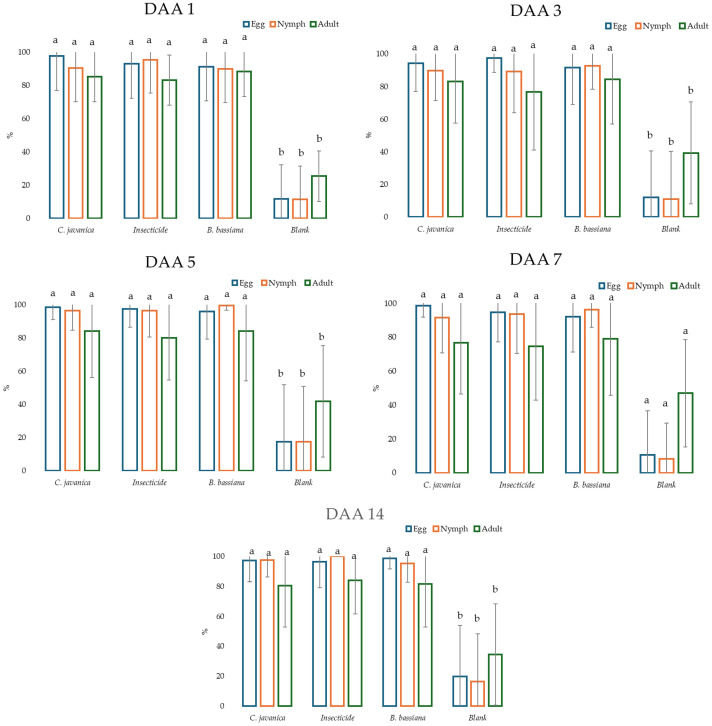
Efficiency of pesticides with respect to day after application (DAA) against eggs, nymphs, and adults of whitefly on eggplants in a greenhouse. Means with the same letter are not significantly different (*p* > 0.05).

## Data Availability

The original contributions presented in this study are included in the article. Further inquiries can be directed to the corresponding author.

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
