# Peer review of "Microbial Biopesticides to Control Whiteflies in Eggplant Solanum melongena, in Greenhouse"

_insects, 2025, doi:10.3390/insects16060578_

Round 1
Reviewer 1 Report
Comments and Suggestions for Authors
The authors have done an interesting job addressing such a current topic as the replacement of synthetic insecticides with biopesticides that are environmentally friendly and safer for both workers and consumers of vegetables. However, my main concern regarding this manuscript is the lack of detail in the methodology used and some omissions that I believe should be included. Below, I outline my observations and suggestions.
Firstly, the authors do not mention whether the taxonomic identification of T. vaporariorum was carried out, nor how this was done. This is of critical importance, as the introduction mentions that both T. vaporariorum and Bemisia tabaci are common species in Mexico. I understand that, without prior identification, the title should be modified to refer directly to whiteflies, rather than just Trialeurodes.
The authors also do not mention whether an initial sampling was conducted to assess the pest density in each of the blocks to be treated. Was the population naturally colonizing the eggplant crop? When did the application begin? Was the intervention threshold (number of eggs, nymphs, and/or adults per leaf) considered?
Additionally, the manuscript does not describe the distance between the blocks that received the different treatments, nor the volume of commercial insecticide/mycoinsecticide applied to each block or plant, nor whether any precautions were taken to prevent the treatments from affecting each other.
Finally, in Figure 1, significant differences are observed between the commercial insecticide and the mycoinsecticides, with the latter showing greater control. However, this information is not presented as a result in the manuscript text. Furthermore, the letter "b" is missing above the control bar in this figure.
In conclusion, based on my review, I do not recommend the manuscript for publication in its current state. I encourage the authors to consider my observations and suggestions, which are intended to improve the quality of the manuscript.
Author Response
Comments1: Firstly, the authors do not mention whether the taxonomic identification of T. vaporariorum was carried out, nor how this was done. This is of critical importance, as the introduction mentions that both T. vaporariorum and Bemisia tabaci are common species in Mexico. I understand that, without prior identification, the title should be modified to refer directly to whiteflies, rather than just Trialeurodes.
Response 1: Only the term whitefly is mentioned (2,14,22, 25, 29, 34, 76, 209, 211, 212)
Comments 2: The authors also do not mention whether an initial sampling was conducted to assess the pest density in each of the blocks to be treated.
Response 2: An initial sampling was carried out prior to the application of the biopesticides in each experimental plot, which was carried out at the beginning of the day. (109)
Comments 3: Was the population naturally colonizing the eggplant crop?
Response 3: We worked with natural populations that invaded the eggplant crop (119)
Comments 4: When did the application begin?
Response 4: The application was carried out in the afternoon, the same day that the initial sampling was carried out (111)
Comments 5: Was the intervention threshold (number of eggs, nymphs, and/or adults per leaf) considered?
Response 5 Continuous monitoring was carried out to know the number of plants with the presence of adults, nymphs and eggs. When 60% of infested plants were registered, it was decided to apply biopesticides. (121)
Comments 6: Additionally, the manuscript does not describe the distance between the blocks that received the different treatments,
Response 6: Each experimental plot had 8 plants, (two rows of four plants each) the two central plants were the sampling unit. (96)
Comments 7: nor the volume of commercial insecticide/mycoinsecticide applied to each block or plant,
Response 7: 80 mL of mixture was applied per experimental plot. (124)
Comments 8: nor whether any precautions were taken to prevent the treatments from affecting each other.
Response 8: For the application, each plot was surrounded by a plastic barrier for isolation. (123)
Comments 9: Finally, in Figure 1, significant differences are observed between the commercial insecticide and the mycoinsecticides, with the latter showing greater control. However, this information is not presented as a result in the manuscript text. Furthermore, the letter "b" is missing above the control bar in this figure.
Response 9: It was added the following text: Figure 2 shows significant differences between commercial insecticide and mycoinsecticides, with the latter showing greater control. However, this information is not presented as a result in the text of the manuscript. (186)
Reviewer 2 Report
Comments and Suggestions for Authors
Cabrera-Mireles et al. evaluate the efficacy of two entomopathogenic fungi, Cordyceps javanica and Beauveria bassiana, against the greenhouse whitefly Trialeurodes vaporariorum on eggplant and compare these with a commercial Spirotetramat treatment and an untreated control. The study's strengths are a fully randomized design with 16 replicates per treatment, direct observation of all three developmental stages (eggs, nymphs, adults) across five sampling dates, and application of the Henderson–Tilton formula for estimation of efficacy. Weaknesses are a lack of baseline levels of infestation, lack of environmental monitoring (e.g., temperature, humidity) that are essential to fungal performance, and little reporting of statistical data beyond non-parametric Kruskal–Wallis tests. Overall quality is rated 72/100, and language quality is 7/10.
The Methods section describes treatment applications (2.5 mL L⁻¹ at 1×10⁸ spores mL⁻¹ for biopesticides; 1.25 mL L⁻¹ for Spirotetramat), pH adjustment and adjuvant application, and afternoon spray timing. Plot size, plastic mulch, and cover material are detailed, but critical abiotic parameters are omitted. Standardized 1.44 cm² counts and two leaves per plant were employed for sampling eggs and nymphs, but there is no documentation of sampling accuracy verification or conidial viability. Statistical analysis employed the Kruskal–Wallis test at each days-after-application (DAA) separately without adjustment for multiple comparisons or effect size reporting.
The Discussion strongly compares current results (93–97% egg mortality; 73–89% adult mortality) to a broad range of literature, yet remains descriptive with no mechanistic explanations for why C. javanica worked better than past reports or how fungal persistence varies under greenhouse conditions. Cost-benefit analysis, potential non-target effects, and regulatory pathways for microbial biopesticides are not addressed, limiting wider application of the results.
Some of the most urgent and challenging points of the work are the following remarks:
Single use of one fungal concentration and commercial dose disallows determination of dose–response relationships necessary to establish optimal field rates.
No environmental monitoring (temperature, relative humidity, UV exposure) compromises reproducibility and interpretation of fungal efficacy since entomopathogen action is extremely susceptible to these factors.
Pre- and post-treatment infestation levels are not given in their original form and, as such, it is impossible to determine the size of treatment effect or pre-existing population heterogeneity.
Statistical analyses are not corrected for multiple comparisons of DAA and do not contain presentation of confidence intervals or medians, heightening suspicions of inflated Type I error and unclear treatment rankings.
No measurement of fungal viability (i.e., spore germination assays) or colony forming units was conducted to confirm active inoculum at time of application.
Potential confounding effects of pH modification and adjuvant addition on mortality of insects are not controlled by adjuvant‑alone treatments.
No information is available on non‑target organisms (i.e., useful predators, pollinators) and duration of residue persistence, limiting evaluation of environmental safety.
Greenhouse conditions (e.g., ventilation rate, light intensity) are not properly described to allow replication in other centers.
Minor other problems are:
Inconsistent nomenclature: Figure 1's "Nimph" must be "Nymph".
The abbreviation DAA is defined late; consider the definition in the first mention in the Abstract.
Figures do not show error bars or raw data tables, concealing variability and replicability.
References to software ("Statistica 7.0") do not mention version publication dates or used settings.
Methodology text at times shifts between past and present tense, losing clarity.
Inconsequential typographical mistakes (e.g., absence of space in "GreenhouseApplications").
Overall impressions of reported results are that C. javanica and B. bassiana are both likely to reach control levels as great as Spirotetramat under tested greenhouse conditions and hence qualify as good prospects for use in sustainable whitefly management programs.
Recommend to editor: Accept with major revisions to increase experimental rigor—particularly environmental control, statistical reporting, and mechanistic discussion—before publication.
Author Response
Comment 1. The Methods section describes treatment applications (2.5 mL L⁻¹ at 1×10⁸ spores mL⁻¹ for biopesticides; 1.25 mL L⁻¹ for Spirotetramat), pH adjustment and adjuvant application, and afternoon spray timing. Plot size, plastic mulch, and cover material are detailed, but critical abiotic parameters are omitted.
Response 1. Temperature and relative humidity were recorded during the study period (Line 88)
The average temperature recorded during the study period was 27.5± 0.26 ºC with a maximum value of 37.9 and a minimum of 21.8 ºC and the relative humidity was 69.4±0.53% with a maximum value of 85.6 and a minimum of 59.4%. (Lines 145-148).
Comment 2. Standardized 1.44 cm² counts and two leaves per plant were employed for sampling eggs and nymphs, but there is no documentation of sampling accuracy verification or conidial viability.
Response 2. Random samples were taken from the whiteflies killed, which were stocked in agar dextrose saboraud culture medium and incubated at 25 ºC, to confirm that they were infected by the applied entomopathogenic fungi. I consider evaluating the viability of the conidia of infected organisms, which is irrelevant for the purposes of the study. (Lines 210-213)
Comment 3. Statistical analysis employed the Kruskal–Wallis test at each days-after-application (DAA) separately without adjustment for multiple comparisons or effect size reporting.
Response 3. The following text was added:
and the multiple comparison of the medians of the ranges of groups with a significance level of 5% was also selected, as described by Siegel & Castellan (1988) and the electronic manual of Statistica software. (Lines 136-138)
Comment 4. The Discussion strongly compares current results (93–97% egg mortality; 73–89% adult mortality) to a broad range of literature, yet remains descriptive with no mechanistic explanations for why C. javanica worked better than past reports or how fungal persistence varies under greenhouse conditions.
Response 4. In this study, experimental data demonstrate that the products evaluated in their current formulation are efficient in controlling the white fly pest. This knowledge is unprecedented and opens up great possibilities for its mass application. This is the first report of their efficiency against whiteflies in vegetables and it was shown that they are as effective as the best insecticide recommended against whiteflies in Mexico. Probably, in a subsequent study we will seek to determine the optimal concentration, but in the meantime, in its current formulation it can already be recommended, both entomopathogens are of national origin, obtained in national collections, which gives them an added value for their probable better adaptation to our environments, but this should be part of a subsequent study. (Lines 275-284)
Comment 5. Cost-benefit analysis, potential non-target effects, and regulatory pathways for microbial biopesticides are not addressed, limiting wider application of the results.
Response 5. The conditions that can affect the results in effectiveness in the different evaluations may be due to various factors, both physical such as temperature and radiation, as well as production factors, since each batch of production in the laboratory may have variations that affect its effectiveness, however, the use of the blank eliminates these variations
For the regulation, these evaluations are the first step to conclude in a sanitary registration, the cost benefit is manifested in the results comparable to the reference chemical, together with the environmental benefits, so it is difficult to establish a monetary value.
The effect on non-target populations would be the subject of another study. It is important to mention that the products evaluated are already in use against different pests in different environments.
Some of the most urgent and challenging points of the work are the following remarks:
Comment 6. Single use of one fungal concentration and commercial dose disallows determination of dose–response relationships necessary to establish optimal field rates.
Response 6. This is not a study to estimate lethal concentration, it is a biological efficiency study of biological products that are already formulated and in use with other pests. In this study, the recommended doses for each product are being used.
Comment 7. No environmental monitoring (temperature, relative humidity, UV exposure) compromises reproducibility and interpretation of fungal efficacy since entomopathogen action is extremely susceptible to these factors.
Response 7. The average temperature recorded during the study period was 27.5± 0.26 ºC with a maximum value of 37.9 and a minimum of 21.8 ºC and the relative humidity was 69.4±0.53% with a maximum value of 85.6 and a minimum of 59.4%.
Comment 8. Pre- and post-treatment infestation levels are not given in their original form and, as such, it is impossible to determine the size of treatment effect or pre-existing population heterogeneity.
Response 8. Graphs of the average number of individuals sampled on all dates were added, including the initial sampling that was carried out before the application of the treatments. It was observed that the populations in the plots of the Blanco sampled numbers tend to increase in sampling 1 and subsequently always fluctuate in greater numbers than that recorded in the plots of pesticides. (Lines 140-179).
Comment 9. Statistical analyses are not corrected for multiple comparisons of DAA and do not contain presentation of confidence intervals or medians, heightening suspicions of inflated Type I error and unclear treatment rankings.
Response 9. Standard deviation values for each treatment were aggregated in each of the outcome graphs. (Lines 189-205)
Comment 10. No measurement of fungal viability (i.e., spore germination assays) or colony forming units was conducted to confirm active inoculum at time of application.
Response 10. If the corresponding measurements of the number of spores and their viability in the laboratory were carried out, for each of the treatments performed, since they are essential to carry out the formulation and adjustment of the dose. The entomopathogens evaluated were verified to contain 80% viability.
Comment 11. Potential confounding effects of pH modification and adjuvant addition on mortality of insects are not controlled by adjuvant‑alone treatments.
Response 11. The procedure for the preparation of entomopathogenic fungi corresponds to the standardized procedure in Mexico.
Comment 12. No information is available on non‑target organisms (i.e., useful predators, pollinators) and duration of residue persistence, limiting evaluation of environmental safety.
Response 12. This topic is not part of this work. Possibly it will be part of another studio. Currently these entomopathogenic fungi are being used against other pests in various environments in Mexico.
Comment 13. Greenhouse conditions (e.g., ventilation rate, light intensity) are not properly described to allow replication in other centers.
Response 13. We only recorded temperature and humidity conditions, as they are the critical factors for the success of entomopathogenic fungi
Minor other problems are:
Comment 14. Inconsistent nomenclature: Figure 1's "Nimph" must be "Nymph".
Response 14. It has already been corrected in the text
Comment 15. The abbreviation DAA is defined late; consider the definition in the first mention in the Abstract.
Response 15. It has already been corrected in the text
Figures do not show error bars or raw data tables, concealing variability and replicability.
Comment 16. References to software ("Statistica 7.0") do not mention version publication dates or used settings.
Response 16. It has already been corrected in the text
Comment 17. Methodology text at times shifts between past and present tense, losing clarity.
Response 17. It has already been corrected in the text
Comment 18. Inconsequential typographical mistakes (e.g., absence of space in "GreenhouseApplications").
Response 18. It has already been corrected in the text.
Round 2
Reviewer 1 Report
Comments and Suggestions for Authors
Authors have incorporated all my suggestions. I think manuscript has improved and may be considered for publication.
Author Response
I have not received any comment from editor 1 round 2
Reviewer 2 Report
Comments and Suggestions for Authors
The authors have labored diligently to address some of the issues of substance raised in the initial round of review, but some underlying deficiencies still require correction before the manuscript is acceptable for publication. In Methods, authors have here presented the greenhouse temperature and relative‐humidity regimes with a mean temperature of 27.5 ± 0.26 °C (21.8–37.9 °C) and a mean relative humidity of 69.4 ± 0.53 % (59.4–85.6 %), thereby meeting the requirements for abiotic critical parameters. They have also mentioned that two leaves per plant and standardized area measurements (1.44 cm²) were used and detailed re‐isolation of fungi from cadavers on agar, which established infection. However, the manuscript still lacks in the form of a formal assessment of conidial viability at the time of application; the authors reference 80 % viability in their response but never detail the germination assay or present those data.
In statistical practice, supplementing with a post-hoc multiple-comparison of medians (Siegel & Castellan, 1988) affords some Type I error control, but the fact that there are no effect-size indices obscures how substantial differences are. Again, figures continue to lack error bars or interquartile ranges, and the use of only letters to compare significance conceals variability. To allow for transparency and replicability, authors should report effect sizes (e.g., nonparametric η² or Cliff's Δ), graph error bars (± SD or 95 % CI), and, if possible, report median values with ranges or quartiles.
In the Discussion, novelty and possible mass application of Cordyceps javanica–based biopesticides are emphasized, but no mechanistic rationale for why their formulation is better than in earlier reports is given. Even a superficial exploration of fungal traits—e.g., germination rate, cuticle–penetration enzymes, UV tolerance in greenhouse light regimes, or survival in plastic‐mulched situations—would strengthen the interpretation. Relatedly, other critical greenhouse factors (UV radiation flux, light intensity, ventilation rates) remain unreported, making replication impossible and limiting the broader applicability of the findings.
Critical practical considerations remain unexplored as well. Though the authors state that their biopesticides provide mortality levels comparable to chemical control, they do not offer any quantitative cost–benefit analysis or regulatory options under Mexican registration programs. Non-target effects (against predators and pollinators) and residue persistence, although left for future investigation, must at least be afforded brief contextualization to enable readers to assess environmental safety. Also, the lack of an adjuvant-only control leaves open the possibility that pH adjustment or surfactants are causing insect death independent of the action of the fungal spores.
To its credit, some minor editorial problems like the "Nimph" misspelling, late definition of DAA in the Abstract, tense inconsistencies, and software reference details have been corrected. Nonetheless, in light of the above-discussed existing shortcomings in spore‐viability documentation, statistical transparency, mechanistic interpretation, environmental parameter reporting, and formulation controls, I recommend major revision. Adding these elements will greatly enhance the rigor, reproducibility, and application value of the study.
Author Response
Comment 1. However, the manuscript still lacks in the form of a formal assessment of conidial viability at the time of application; the authors reference 80 % viability in their response but never detail the germination assay or present those data.
Response 1. To estimate the percentage of germination (viability) of the conidia of the biopesticides evaluated, the methodology described by Gómez et al., 2014 was followed; Vélez et al., 1997. Dilutions were prepared in a final volume of 15 mL of 0.1% Inex solution, at a final concentration of approximately 9-12x106 conidia/mL. Subsequently, 5 μL of the prepared dilution was taken and inoculated in Petri dishes with potato, dextrose, agar (PDA) using a sterile angled rod. The inoculated dishes were incubated at 25 +/- 2°C for 24 hours. After this incubation time in the laminar flow chamber, a portion of agar of about 1 cm2 was cut and placed on a slide, a drop of lactophenol blue was added and covered with a coverslip, taking 5 samples per plate, and at least 100 conidia per sample were recorded. counting germinated and non-germinated conidia (germinated conidia were considered those whose germination tube is twice as large as the diameter of the conidia) and finally the average of the 5 readings was obtained and the percentage of germinated and non-germinated conidia was calculated, using the following formula: (a = number of germinated conidia, b = number of ungerminated conidia). Lines 96-133
Commment 2. In statistical practice, supplementing with a post-hoc multiple-comparison of medians (Siegel & Castellan, 1988) affords some Type I error control, but the fact that there are no effect-size indices obscures how substantial differences are. Again, figures continue to lack error bars or interquartile ranges, and the use of only letters to compare significance conceals variability. To allow for transparency and replicability, authors should report effect sizes (e.g., nonparametric η² or Cliff's Δ), graph error bars (± SD or 95 % CI), and, if possible, report median values with ranges or quartiles.
Response 2. This text was added to methodology: The post-hoc tests were complemented with a descriptive analysis by means of a graph of the confidence interval of the mean for each BP performed for the variables Percentage of mortality of flies in the egg, pupal and adult stages, in order to show the marked differences between BPs that caused the significance obtained in the Kruskal-Wallis test. for this reason, it was not necessary to calculate the size of the non-parametric effect when comparing two groups of observations with the Cliff Delta statistic, which does not perform a hypothesis test and only serves to indicate whether two groups or treatments are equal when the value is approximately equal to zero or different if it is equal to 1 or -1 Macbeth, Razumiejczyk & Ledesma (2011). Lines 219-227
shown through mean confidence intervals. Line 234
Comment 3. In the Discussion, novelty and possible mass application of Cordyceps javanica–based biopesticides are emphasized, but no mechanistic rationale for why their formulation is better than in earlier reports is given. Even a superficial exploration of fungal traits—e.g., germination rate, cuticle–penetration enzymes, UV tolerance in greenhouse light regimes, or survival in plastic‐mulched situations—would strengthen the interpretation.
Response 3. I believe that, according to the experimental approach of this study, the comparisons are made with the blank (negative control) and with the insecticide (positive control. And taking into consideration that the fundamental purpose was to determine if the efficiency of the two biopesticides equals or improves on the efficiency of the commercial insecticide that today is the most efficient against the whitefly pest, the results are conclusive in showing that both biopesticides compete efficiently against the synthetic insecticide. This is the first report that offers a viable alternative against whiteflies in greenhouse vegetables.
Comment 4. Relatedly, other critical greenhouse factors (UV radiation flux, light intensity, ventilation rates) remain unreported, making replication impossible and limiting the broader applicability of the findings.
Response 4. Modern vegetable production occurs in greenhouses covered with anti-aphid mesh with a plastic cover under the roof that blocks UV rays 10 to 50 times. This system favors the yield and quality of vegetables and reduces the presence of pests and diseases. Lines 67-70, 140
Comment 5. Critical practical considerations remain unexplored as well. Though the authors state that their biopesticides provide mortality levels comparable to chemical control, they do not offer any quantitative cost–benefit analysis or regulatory options under Mexican registration programs.
Response 5. The cost of the biopesticides applied in this work was $7.5 USD, while the cost of the applied insecticide was $150.0 USD. This shows that just because of the unique cost of each product, biopesticides offer greater economy. They are cheaper in long run, show lesser residual effects, and are able to overcome the problem of resistance. Lines 78-79
Comment 6. Non-target effects (against predators and pollinators) and residue persistence, although left for future investigation, must at least be afforded brief contextualization to enable readers to assess environmental safety.
Response 6. Among microbial control agents, most experiments on non-target organisms resulted in overall compatibility with biological control agents, but some results were controversial (Lisi et al., 2025), on the other hand, Abonyo et al., 2016 reported that when evaluating M. anisopliae they did not observe direct effects on beneficial fauna. Lines 79-82
Comment 7. Also, the lack of an adjuvant-only control leaves open the possibility that pH adjustment or surfactants are causing insect death independent of the action of the fungal spores.
Response 7. The Inex adjuvant is widely used and is one of those recommended to avoid any negative effect on entomopathogenic fungi. This product is not recommended for its insecticidal action, rather it is harmless. As it is a harmless product, there is no possibility of it exerting any insecticidal action. On the other hand, the Inex was added in the same concentration in the biopesticides and the insecticide, so if there was any collaboration as an insecticide, this action would be the same for the three treatments, so its possible action would occur in the same way. In conclusion, the presence of Inex cannot be considered as a possible source of variability in the experiment.

Round 3
Reviewer 2 Report
Comments and Suggestions for Authors
In this new edition, the authors have provided a detailed germination assay protocol—complete with sampling methodology, incubation conditions, and % viability calculations—to underpin their assertion of 80 % conidial viability at application time (Comment 1). They have also clarified the greenhouse environment, referring to use of anti-aphid mesh and UV-blocking plastic covering (Comment 4), and have provided terse context to non-target safety by referring to compatibility studies with beneficial fauna (Comment 6).
There remain some concerns, though. Although they added mean confidence interval plots to their Kruskal–Wallis output, authors still fail to report formal nonparametric measures of effect size (e.g., η² or Cliff's Δ) and have yet to report median values with interquartile ranges or refigured figure panels to contain accurate error bars (Comment 2). The Discussion continues to lack any mechanistic rationale for the improved performance of Cordyceps javanica preparations (Comment 3), and quantitative cost–benefit comparison or guidance on regulatory pathways under Mexican registration programs has not been offered (Comment 5). Finally, while the authors make a claim that their adjuvant of choice (Inex) is inert, they have neither explained this in terms of reference to toxicity studies nor provided an adjuvant-only control treatment (Comment 7).
Given these few outstanding issues—most significantly the absence of effect-size reporting, mechanistic explanation, economic analysis, and an adjuvant control—I recommend a decision of minor revision. The manuscript has been improved in several areas but would vastly gain from the authors' resolution of the existing methodologic and interpretive limitations before being considered for publication.
Author Response
Round 3 Responses.
Comment 1.
There remain some concerns, though. Although they added mean confidence interval plots to their Kruskal–Wallis output, authors still fail to report formal nonparametric measures of effect size (e.g., η² or Cliff's Δ) (Comment 2).
Response 1.
The size of the non-parametric effect when comparing two groups of observations with the Cliff Delta statistic is not calculated by the statistical software SAS, and SPSS do not calculate the Dominance analysis of the Cliff Delta statistic for ordinal scale variables (Feng and Cliff, 2009) and the same happens with other statistical software such as STATISTICA (version 7) and InfoStat. When reviewing the Syllabus of the course of "Statistical Methods" in the Non-parametric Statistics Unit, this technique of Cliff's Delta calculation is not taught, which had its development in psychological research, but in agriculture, based on the review of 64 articles evaluating insecticides in vegetable crops, only one used Kruskal-Wallis with Dunn's median comparison test and none performed these complementary studies (Size Measurement of the [EMS] effect) to the post-hoc comparison of medians of the treatments, as requested by the editor 2 round 2. With respect to the program made by Macbeth, Razumiejczyk & Ledesma (2011) in the Visual Basic 6.0 programming language has the limitation of forming a data matrix with 30 lines and 30 columns. On the other hand, the program carried out in Fortran by Feng, Du and Cliff, Normal (2009), performs all the estimator and confidence interval calculations for Delta when there are up to 75 data or lines, in case it is exceeded, it only calculates the Delta statistic and its components appear in the results. As it is in my work.
The bibliographic references that support what is referred to in APPENDIX are presented on the attached page
Comment 2.
The Discussion continues to lack any mechanistic rationale for the improved performance of Cordyceps javanica preparations (Comment 3), and quantitative cost–benefit comparison or guidance on regulatory pathways under Mexican registration programs has not been offered (Comment 5).
Response 2. regulatory pathways under Mexican registration programs
This work was planned in accordance with Mexican regulations that establish the regulatory procedures to achieve the legal recognition of a product with a biopesticide effect. This work coincides with the initial part to access this regulation and consists of carrying out the biological efficiency test, which must demonstrate that a biopesticide (or any pesticide in general) has control effects on a pest organism. The regulations in force for Mexico are contained in the Official Mexican Standard NOM-032-FITO-1995, which establishes the phytosanitary requirements and specifications for the performance of studies of the biological effectiveness of agricultural pesticides and their technical opinion, as well as the DECREE by which various provisions of the Regulations on Registrations are reformed, added and revoked. Import and export authorisations and export certificates for pesticides, plant nutrients and toxic or dangerous substances and materials. This decree defines a microbial pesticide as a product formulated based on microorganisms, such as bacteria, viruses, fungi, yeasts, nematodes or protozoa, used for pest control purposes; Article 7.- The products whose registration will be subject to the procedure provided for in these Regulations are classified as: Microbial pesticides for agricultural, domestic, forestry, gardening, livestock, public health and urban use (Bacteria; Fungi; Virus; Nematodes; protozoa and algae).
Quantitative cost–benefit comparison is out of the scope of this work. This issue should be addressed in future works leading to propose these mycoinsecticides to public use. Meanwhile it is not considered.
Comment 3.
Finally, while the authors make a claim that their adjuvant of choice (Inex) is inert, they have neither explained this in terms of reference to toxicity studies nor provided an adjuvant-only control treatment (Comment 7).
Response 3.
- I only found one article (Zamora-Avilés el al., 2022) that shows that Inex caused the loss of viability of the spores; in this work, the entomopathogenic fungus was stored in a mixture of 0.5% Inex in water and the evaluation was carried out after 30 days, which is completely inadequate, since entomopathogenic fungi are not stored by adding Inex.
Zamora-Avilés, N., Orozco-Flores, A. A., Gomez-Flores, R., Domínguez-Gámez, M., Rodríguez-Pérez, M. A., & Tamez-Guerra, P. (2022). Increased Attraction and Stability of Beauveria bassiana-Formulated Microgranules for Aedes aegypti Biocontrol. Journal of Fungi, 8(8), 828. https://doi.org/10.3390/jof8080828
- Inex surfactant is used for the purposes of a correct dispersion of conidia in a water aliquot to perform the evaluation of the concentration of spores in a formulation, both in evaluations in Petri dishes and in mixtures that will be immediately applied in pathogenicity or virulence evaluations, as well as for field evaluations. In the various articles consulted, the viability evaluations of strains in mixture with Inex (recommended dose of 0.2 to 0.5%) obtained a viability that generally exceeds 90%. Inex surfactant is commonly used in the formulation of mycoinsecticides in México, Colombia and other parts of the world and has been proven not to affect conidia germination. This product has been used as a surfactant with strains of Beauveria bassiana, Coryceps javanica, Metarhizium anisopliae, Lecanicillium lecanii among others.
We did not find any reason that led us to include Inex as a control treatment. In addition, as stated above, Inex was added equally in all three pesticides evaluated, so it cannot be a factor of experimental variability. Below, are bibliographic references where they support with experimental work that Inex does not affect the viability of the spores:
Cortez-Madrigal, Hipolito. (2025). Evaluación de aislamientos de hongos entomopatógenos y su virulencia hacia Bactericera cockerelli, según su origen. Fitopatologia Colombiana. 34. 17-21.
Esteban-Del Ángel, R.; Silva-Martínez, K. L. , Allende-Molar R.; Arrieta-González, A.; Silva-Rojas,H. V.; Martínez-Sánchez, I. "Pathogenicity of Metarhizium sp. on Spodoptera frugiperda in Maize in Veracruz‚ Mexico," Southwestern Entomologist, 2025.50(1), 1-19,
Ordaz-Hernández, A., Montesinos-Matías, R., Mellín-Rosas, M.A. et al. Improvement of the production and quality of Cordyceps javanica conidia for the control of Diaphorina citri adults. World J Microbiol Biotechnol 40, 115 (2024). https://doi.org/10.1007/s11274-024-03922-2
Raymundo-Jiménez, Raúl & García-Ibarra, Elías & López-Arroyo, J. Isabel & R.-Cabral, Nadiezhda & Rodriguez Guerra, Raul. (2019). Producción y germinación de conidios del hongo entomopatógeno Hirsutella citriformis (Ascomycota: Ophiocordycipitaceae). Scientia Fungorum. 49. e1221. DOI:10.33885/sf.2019.49.1221
Villegas-Rodríguez, F., Díaz-Gómez, O., Casas-Flores, J. S., Monreal-Vargas, C. T., Tamayo-Mejía, F., & Aguilar-Medel, S. (2017). Activity of two entomopathogenic fungi, molecularly identified, on bactericera cockerelli. Revista Colombiana De Entomología, 43(1), 27–33. https://doi.org/10.25100/socolen.v43i1.6643
Zapata-Narváez, Yimmy Alexander, & Botina-Azain, Blanca Lucia. (2023). Effect of adjuvants, fungicides and insecticides on the growth of Trichoderma koningiopsis Th003. Revista mexicana de fitopatología, 41(3), 412-433. Epub 13 de octubre de 2023. https://doi.org/10.18781/r.mex.fit.2305-1
APPENDIX
Bibliographic references consulted for the comments on statistics that support the writing in document ROUND 3. Response 1. May 18th, 2025
- Abdel-Raheem, M.A. y Al-Keridis L.A (2017) Virulence of Three Entomopathogenic Fungi Against Whitefly, Bemisia tabaci (Gennadius)(Hemiptera: Aleyrodidae) in Tomato Crop. J. Entomology Vol 14(4) pp.155-159
- Abubakar, M., Yadav, D., Koul, B. y Song, M. (2023). Efficacy of Eco-Friendly Bio-pesticides against the whitefly Bemisia Tabaci (Gennadius) for sustainable Eggplant Cultivation in Kebbi state, Nigeria. Agronomy vol. 13
- Ain, Q., Mohsin, A., Naeem, M. y Shabbir, G. (2021). Effect of entomopathogenic fungi, Beavaria bassiana and Metarhizium anisopliae on thrips tabaci lindeman (thysanoptera thripidae) populations in different onion cultivars. Egyptian Journal of Biological pesst control, pp: 31:97.
- Ain, Q., Mohsin, A.U., Naeem, M. et al. (2021). Effect of entomopathogenic fungi, Beauveria bassiana and Metarhizium anisopliae, on Thrips tabaci Lindeman (Thysanoptera: Thripidae) populations in different onion cultivars. Egypt J Biol Pest Control 31, 97. https://doi.org/10.1186/s41938-021-00445-y
- Akmal, M. , Nacem, M. y Tahira, H. (2013) Efficacy of Beauveria Bassiana (Deuteromycotina: Hypomycetes) Against Different Aphid Species Under Laboratory Conditions. Pakistan J. Zool. Vol 45(1) pp. 71-78.
- Alano, D. M., Araujo, E. S., Mirás-Avalos, J. M., Pimentel, I. C., & Zawadneak, M. A. C. (2021). Short communication: Sublethal effects of insecticides used in strawberry on Trichogramma pretiosum (Hymenoptera: Trichogrammatidae). Spanish Journal of Agricultural Research, 19(1), e10SC01. https://doi.org/10.5424/sjar/2021191-17235
- Amano, H. y Hasseb, M. (2001) Recently-proposed methods and concepts of testing the effects of pesticides on the beneficial mite and insect species: limitations and implications in IPM. Appl. Entrmol. Zool, 36(1): 1-11.
- Ananthakrishnan, T. N. (1993) Bionomics of thrips. Annu. Rev. Entomol. Vol 38 pp. 71-92
- Ansari, M., Brownbridge, M., Shah, F. y Butt, T.(2008) Efficacy of entomopathogenic fungi against soil-dwelling life stages of western flower thrips, Frankliniella occidentalis, in plant-growing media. Entomologia Experimentalis et applicata vol 127 pp. 80-87.
- Ashtari, S., Sabahi, Q. y Talebi, K. (2018)) Evaluation of toxicity of some biocompatible insecticides on Trichogramma brassicae and T. evanescens under laboratory and semi-field conditions. J. Crop Prot. 2018, 7(4): 459-469.
- Avery, P., Kumar, V., Skvarch, E., Mannium, C., Povell, C., Mackenzie, C. y Osborne, L.(2019). An Ecological Assesment of Isaria Fumosorosea Applications Compared to a Neocotinoid tratment for regulating invasive ficus whitefly. Journal of Fungi, vol 5(36).
- Bohatá, A., Azeez, E., Lencová, J., Osborne, L. y Mraz, J. (2024). Control of sweet potato whitefly (Bemisia tabaci) using entomopathogenic fungi under optimal and suboptimal relative humidity conditions. Pest management Science, vol. 80, pp: 1065-1075.
- Canassa, F., Tall, S., Moral, R., Lara, I., Delalibera, I y Meyling, N. (2019) Effects of bean seed treatment by the entomopathogenic fungi metarhizium robertsii and beauveria bassiana on plant growth, spider mite populations and behavior of predatory mites. Biological Control Vol. 132 pp. 199-208
- Castillo-Ramirez, O., Guzman-Franco, A., Santillán-Galicia, M. y Tamayo-Mejia, F.(2020) Interaction between predatory mites( Acari: Phytoseiidae) and entomopathogenic fungi in Tetranychus urticae populations. BioControl Vol 65 pp. 433-445.
- Castro, T. Eilenberg, J. y Delilbera, I. (2018) Exploring virulence of new and less studied species of Metarhizium spp. From Brazil for two-sportted spider mite control. Exp. Appl. Acarol Vol 74 pp. 139-146.
- Chandler, D. Davidson, G. y Jacobson R. (2005) Laboratoy and glasshouse evaluation of entomopathogenic fungi against the two-spotted spider mite tetranychus urticae (Acari- tetranychidae) on tomato, lycopersicon esculentum. Biocontrol Science and Technology Vol. 15(1) pp. 37-54
- Choudhary, R. S., Rana, B. S., Mahla, M.K. y Meena, A. K. (2018) Bioefficacy of Biorational Insecticides against Larval Population of leucinodes orbonalis (Guen.) in Brinjal. International Journal of Current Microbiology and Applied Sciences, vol 7(7), pp: 47-60
- Chouikhi, S., Assadi, B.H., Lebdi, K.G. et al. (2022) Efficacy of the entomopathogenic fungus, Beauveria bassiana and Lecanicillium muscarium against two main pests, Bemisia tabaci(Genn.) and Tetranychus urticae (Koch), under geothermal greenhouses of Southern Tunisia. Egypt J Biol Pest Control 32, 125. https://doi.org/10.1186/s41938-022-00627-2
- Cuthbertson, A. y Walters K. (2005) Pathogenicity of the entomopathogenic fungus, lecanicillium muscarium, againts the sweetpotato whitefly Bemisa tabaci under laboratory and glasshouse conditions. Mycopathologia Vol. 160 pp- 315-319
- Dadther-Huaman, H., Machaca-Paccara, A. y Quispe-Castro, R. (2020) Eficacia de nueve métodos de control de Oregmapyga peruviana (Granara de Willink & Diaz) (Hemiptera coccoidea: Eriococcidae) en vitis vinifera L. "negra criolla" y "quebranta". Scienia Agropecuaria, vol 11(1), pp: 95-103.
- De Barro, P., Liu, S. Boykin L. y Dinsdale, A. (2011) Bemisia Tabaci: A statement of Species Status. Annu. Rev. Entomol. Vol 56 pp 1-19.
- Elhakim, E., Mohamed, O. y Elazouni, I. (2020) Virulence and proteolytic activity of entomopathogenic fungi against the two-spotted sipder mite, Tetranychus urticae koch (Acari; Tetranychidae). Egyptian Journal of Biological Pest Control
- Farazmand, A. y Amir-Maafi, M. (2018) A population growth model of Tetranychus urticae Koch (Acari: Tetranychidae). Persian J. Acarol. Vol 7(2) pp. 193-201
- Faria, M. y Wraight, S. (2007) Mycoinsecticides and Mycoacaricides: A comprehensive list with worlwide coverage and international classification of formulation types. Biological Control Vol. 43 pp. 237-256
- Feng, Du and Cliff, Normal (2009) "JMASM29: Dominance Analysis of Independent Data (Fortran)," Journal of Modern Applied Statistical Methods: Vol. 8: Iss. 2, Article 32. Available at: http://digitalcommons.wayne.edu/jmasm/vol8/iss2/32
- Ganga P.N. y Krishnamoorthy A. (2012) Comparative Field Efficacy of Various Entomophathogenic Fungi againts Thrips tabaci: Prospects for Organic Production of Onion in India. Acta Horticulture Vol 933. pp. 433-437.
- Gangai, S., Siddhapara, M. y Shinde, C. (2023) Toxicity of insecticides against egg parasitoid, Trichogrammatoidea bactrae Nagaraja under laboratory conditions. J. ent. Res., 47(4): 694-699. DOI: 10.5958/0974-4576.2023.00127.5
- Gebremariam, A., Chekol, Y. y Assefa, F. (2022). Extracellular enzyme activity of entomo pathoganic fungi, Beauveria baussiana and metarhizium anisopliae and their pathogenicity potential as a Bio-Control agent against whitefly pest, Bemisia tabacci and Trialeurodes Vaporarioum (Hemiptera: Aleyrodidae). BMC Research Notes, vol. 15.
- Gent, D. y Schwartz, H.(2004) Distribution and incidence of iris yellow spot virus in Colorado and Its Relation to Onion Plant Population and Yield. Plant Disease Vol. 88(5) pp. 446-452
- Ghelani, M.K., Kabaria, B.B. y Chhodavadia, S.K.(2014) Field efficacy of various insecticides against major sucking pest of bt cotton. J. Biopest Vol. 7 pp. 27-32
- Ghongade, D.S., Sangha, K.S. Efficacy of biopesticides against the whitefly, Bemisia tabaci (Gennadius) (Hemiptera: Aleyrodidae), on parthenocarpic cucumber grown under protected environment in India. Egypt J Biol Pest Control 31, 19 (2021). https://doi.org/10.1186/s41938-021-00365-x
- Gupta, J. K., Bhatnagar, A. y Agrawal, V. K. (2021) Effectiveness of Bio-rationales and newer pesticides againts damage due to yellow mite, Polyphagastarsonemus latus (Banks) on caosicum (capsicum annuml) under shade net house durring summer. Journal of entomology and zoology studies, vol 9(1), pp: 1989-1993.
- Hamrouni, B., Chouikhi, S., Ettaib, R., Boughalleb, N. y Sadok, M. (2021) Effect of the native strain of the predator nesidiocoris tenuis Reuter and the entomopathogenic fungi Beaveria bassiana and Lecanicillium muscarium againt Bemisia tabaci (Genn.) under greenhouse conditions in Tunisia. Assadi et al Egyptian Journal of Biological Pest Control
- Hamrouni, B., Chouikhi, S., Ettaib, R., Boughalleb, N. y Sadok, M. (2021) Effect of the native strain of the predator nesidiocoris tenuis Reuter and the entomopathogenic fungi Beaveria bassiana and Lecanicillium muscarium againt Bemisia tabaci (Genn.) under greenhouse conditions in Tunisia. Assadi et al Egyptian Journal of Biological Pest Control
- Hamrouni, B., Chouikhi, S., Ettaib, R., Boughalleb, N. y Sadok, M. (2021) Effect of the native strain of the predator nesidiocoris tenuis Reuter and the entomopathogenic fungi Beaveria bassiana and Lecanicillium muscarium againt Bemisia tabaci (Genn.) under greenhouse conditions in Tunisia. Assadi et al Egyptian Journal of Biological Pest Control
- Hamrouni, B., Chouikhi, S., Ettaib, R., Boughalleb, N. y Sadok, M. (2021) Effect of the native strain of the predator nesidiocoris tenuis Reuter and the entomopathogenic fungi Beaveria bassiana and Lecanicillium muscarium againt Bemisia tabaci (Genn.) under greenhouse conditions in Tunisia. Assadi et al Egyptian Journal of Biological Pest Control
- Hassan, D., Rizk, M., Sobhy, H., Mikhail, W. y Nada, M.(2017) Virulent Entomopathogenic Fungi against the two-spotted spider mite tetranychus urticae and some associated predator mites as non target oganisms. Egyptian Academic Journal of Biological Sciences Vol.10(6) pp.37-56
- Havey, M., Hunsaker, D. y Munaiz, E. (2021) Genetic Analysis of the unique epicuticular wax profile of Odouless Greenleaf Onion. J. Amer. Soc. Hort. Sci. Vol. 146(2) pp. 118-124
- Hesketh, H., Alderson, P., Pye, B. y Pell, J. (2008) The development and multiple uses of a standardised bioassay method to selecty hypococrealean fungi for biological control of aphids. Biological Control Vol 46 pp. 242-255
- Jones, David (2003) Plant viruses transmitted by whiteflies. European Journal of Plant Pathology Vol. 109 pp. 195-219
- Khan, I., Shah, A. y Said, F. (2015) Distribution and population dynamics of thrips tabaci(Thysanoptera: Thripidae) in selected districts of Khyber Pakhtunkhwa province Pakistan. Journal of Entomology and Zoology Studies, 3(5): 153-157
- Khan, M., Khan, H. y Ruberson, J. (2015) Lethal and behavioral effects of selected novel pesticides on adults of Trichogramma pretiosum (Hymenoptera: Trichogrammatidae). Pest Manag Sci. DOI: 10.1002/ps.3972.
- Khoury, C., Guillot, J. y Nemer, N. (2020) Susceptibility and development of resistance of the mite Tetranychus urticae to aerial conidia and blastospores of the entomopathogenic fungus Beauveria Bassiana. Systematic & Applied Acarology Vol. 25(3) pp. 429-443
- Kumar, S., Monga, D., Hiremani, N., Nagrale, D., Kranthi, S., Kumar, R., Kranthi, K., Tuteja, O. y Waghmare, V. (2021) Evaluation of bioefficacy potential of entomopathogenic fungi against the whitefly (bemisia tabaci Genn.) on cotton under polyhouse and field conditions. Journal of invertebrate pathology vol. 183
- Liu, T. y Stansly F. (2000) Insecticidal activity of surfactants and oils against silverleaf whitefly (Bemisia argentifolii) nymphs (Homoptera: Aleyrodidae) on collards and tomato. Pest Manag. Sci. Vol. 56 pp. 861-866
- Maniana, N., Bugeme, D., Wekesa, V., Delalibera, I y Knapp, M. (2008) Role of entomopathogenic fungi control of tetranychus evansi and tetranychus urticae (Acari: tetranuchidae), pest of horticultural crops. Exp. Appl. Acarol
- Meza Rodríguez, A. R. . (2021). MÉTODOS ALTERNATIVOS ANTE LA VIOLACIÓN DE SUPUESTOS EN DISEÑOS DE EXPERIMENTOS FACTORIALES. Anales Científicos, 82(2), 318-335. https://doi.org/10.21704/ac.v82i2.1795
- Moura, G., Kobon, N., Dias, E. y Delalibera, I. (2013). The virulence of entomopathogenic fungi against Bemisia Tabacci biotype B (Hemiptera: Aleyrodidae) and ther conidal production using solid substrate fermentation. Biological control, vol. 66 pp: 209-218.
- Mugisho, D., Knapp, M., Iddi, H., Kibira, A., Kalemba, N.(2009) Influence of temperature on virulence of fungal isolates of mertahizium anisopliae and Beaveria bassiana to the Two-spotted spider mite Tetranychus urticae. Mycopathologia Vol 167 pp. 221-227
- Navas-Castillo, J., Fiallo-Olivé, E. y Campos-Sánchez, S. (2011) Emergin Virus Diseases Trasmitted by Whiteflies. Annu. Rev. Phytopathol Vol. 49 pp. 219-248
- Norhelina, L., Sajap, A.S., Mansour, S.A. e Idris, A.B (2013) Infectivity of Five Metarhizium anisopliae (deuteromycota: Hyphomycetales) Strains on Whitefly, Bemisia tabaci (Homoptera:Aleyrodidae) Infesting Brinjal, Solanum melongena (Solanaceae). Academic Journal of Entomology Vol. 6(3) pp. 127-132
- Nozad-Bonab, Z., Khani, S., Maroofpour, N. e Iranipour, S. (2021) Resiudal toxicity of some insecticides on Tuta absoluta (Lepidoptera: Gelechiidae) larvae under laboratory conditions. Journal of entomological society of Iran, 41(2):163-174.
- Nyasani, J. Subrsmanian, S., Poehling, H., Maniana, N., Ekesi, S. y Meyhofer, R. (2015) Optimizing western flower thrips management on Frenc Beans by Combined Use of Beneficals and Imidacloprid. Insects Vol. 6 pp. 279-296.
- Pappas, M., Migkou, F. y Broufas, G.(2013) Incidence of resistance to neonicotinoid insecticides in greenhouse populations of the whitefly, trialeurodes vaporariorum (Hemíptera: Aleyrodidae) from Greece. Appl. Entomol Zool Vol. 48 pp. 373-378
- Park, H. y Kim, K. (2010) Selection of Lecanicillium Strains with High Virulence against Developmental Stages of Bemisia tabaci. Mycobiology Vol. 38(3) pp. 210-214
- Park, Y. y Lee, J. (2002) Leaf Cell and Tissue Damage of Cucumber Caused by Twospotted Spider Mite (Acari: Tetranychidae). J. Econ. Entomol. 95(5): 952-957.
- Poprawski, T., Greenberg, S. y Ciomperlik, M. (2000) Effect of host plant on beaveria bassiana and paecilomyces fumosoroseus induced mortality of trialeurodes vaporariorum (Homoptera: Aleyrodidae). Environmental Entomology Vol 29(5) pp. 1048-1053
- Raheem, M. y Al-Keridis, L. (2017) Virulence of three entomopathogenic Fungi Against Whitefly Bemisia tabaci (Gennadius)(Hemiptera: Aleyrodidae) in tomato crop. Journal of entomology vol.14(4) pp. 155-159
- Sahayaraj, K. y Namasivayam K. (2008) Mass Production of entomopathogenic fungi using agricultural products and by products. African Journal of Biotechnology, Vol.7(12) pp. 1907-1910.
- Saitto T. y Sugiyama K.(2005) Pathogenicity of three Japanese strains of entomopathogenic fungi against the silverleaf whitefly, Bemisia argentifolii. Appl. Entomol. Zool. 40(1): 169-172
- Sayed, D. y Zakaria, H. (2021) Field evaluation of two entomopathogenic fungi; Beuveria bassiana and Metorhizium anisopliae as biocontrol agent against the spiny bollworm, Earias insulana Boisduval (lepidoptera: noctuidae) on cotton plants. Egyptian Journal of Biological Pest Control, pp: 31-78.
- Scorsetti, A., Humber, R., Gregorio, C. y López, C. (2008) New records of entomopathogenic fungi infecting Bemisia tabaci and Trialeurodes vaporariorum, pest of horticultural crops in Argentina. BioControl Vol 53 pp. 787-796.
- Sharma, M., Budha, P. y Pradhan, S. (2015) Efficacy test of bio-pesticides against tobacco whitefly bemisa tabaci(Gennadius, 1889) on Tomato plants in Nepal. Journal of institute of Science and Technology vol. 20(2) pp. 11-17
- Singh, H. y Joshi, N. (2020) Management of the aphid, Myzus persicae (Sulzer) and the whitefly, Bemisia Tabaci (Gennadius), using biorational on capsicum under protected cultivation in India. Egyptian Journal of Biological pest control, pp: 30-67.
- Singh, H., Joshi, N. Management of the aphid, Myzus persicae (Sulzer) and the whitefly, Bemisia tabaci (Gennadius), using biorational on capsicum under protected cultivation in India. Egypt J Biol Pest Control 30, 67 (2020). https://doi.org/10.1186/s41938-020-00266-5
- Topoz, E., Erler, F. y Gumrukcu, E. (2016). Survey of indigenous entomopathogenic fungi and evaluation of ther pathogenicity against the carmine spider mite, tetranychus cinnabarinus (Boisd.), adn the whitefly, Bemisia tabacci (Genn.) Biotype B. Pest Management Science.
- Trdan, S., Vidrih, M. y Rozman L. (2006) Intercropping againts onion thrips, Thrips tabaci Lindeman (Thysanoptera: Thripidae) in onion production: on the suitability of orchard grass, lacy phacelia, and buckwheat as alternatives white clover. Journal of Plant Diseases and Protection Vol 113(1) pp. 24-30.
- Uddin, A., Nazir, T., Abdulle, Y., Hussain, G., Ali, M., Anwar, T., Sokea, T. y Qiu, D. (2020) In vitro pathogenicity of the fungi beaveria bassiana and lecanicillium lecanii at different temperatures against the whitefly, bemisia tobaci (Genn.) (Hemiptera:Aleyrodidae). Keerio et al. Egyptian Journal of Biological Pest Control Vol. 30
- Uddin, A., Nazir, T., Abdulle, Y., Hussain, G., Ali, M., Anwar, T., Sokea, T. y Qiu, D. (2020) In vitro pathogenicity of the fungi beaveria bassiana and lecanicillium lecanii at different temperatures against the whitefly, bemisia tobaci (Genn.) (Hemiptera:Aleyrodidae). Keerio et al. Egyptian Journal of Biological Pest Control Vol. 30
- Uddin, A., Nazir, T., Abdulle, Y., Hussain, G., Ali, M., Anwar, T., Sokea, T. y Qiu, D. (2020) In vitro pathogenicity of the fungi beaveria bassiana and lecanicillium lecanii at different temperatures against the whitefly, bemisia tobaci (Genn.) (Hemiptera:Aleyrodidae). Keerio et al. Egyptian Journal of Biological Pest Control Vol. 30
- Uddin, A., Nazir, T., Abdulle, Y., Hussain, G., Ali, M., Anwar, T., Sokea, T. y Qiu, D. (2020) In vitro pathogenicity of the fungi beaveria bassiana and lecanicillium lecanii at different temperatures against the whitefly, bemisia tobaci (Genn.) (Hemiptera:Aleyrodidae). Keerio et al. Egyptian Journal of Biological Pest Control Vol. 30
- Vega, Fernando (2018) The use of fungal entomopathogens as endophytes in biological control: a review. Mycología Vol. 110(1) pp. 4-30
- Vianna, U., Pratissoli, D., Zanuncio, J., Lima, E., Brunner, J., Pereira, F. y Serrao, J. (2009) Insecticide toxicity to trichogramma pretiosum (Hymenoptera: Trichogrammatidae) females and efect on descendant generation. Ecotoxicology, 18: 180-186 DOI:10.1007/s10646-008-0270-5
- Wawdhane, P. Nandanwar, V., Mahankuda, B., Ingle, A. y Chaple, K. (2020) Bio-efficacy of insecticides and bio pesticides against major sucking pest of Bt-cotton. Journal of Entomology and Zoology Studies, 8(3): 829-833
- Wekesa, V., Maniana, N. Knapp, M. y Boga, H. (2005) Pathogenicity of Beaveria Bassiana and Metarhizium anisopliae to the tobaco spider mite Tetranychus evansi. Experimental and Applied Acarology Vol 36 pp. 41-50
- Wu, S., Sarkar, S.C., Lv, J. et al.(2020) Poor infectivity of Beauveria bassiana to eggs and immatures causes the failure of suppression on Tetranychus urticae population. BioControl 65, 81–90. https://doi.org/10.1007/s10526-019-09970-0
- Wu, s., Xie,H., Li,M., Xu, X. y Lei, Z. (2016) Highly virulent beaveria bassiana strains against the two-spotted spider mite species. Exp. Appl. Acarol. Vol. 70 pp. 421-435
- Zafar, J., Freed, S., Ali, B. y Farooq, M. (2016) Effictiveness of Beauveria bassiana Against Cotton Whitefly, Bemisia tabaci (Gennadius) (Aleyrodidae: Homoptera) on Diferent Host Plants. Pakistan J. Zool. Vol48(1) pp. 91-99